

# Energy consumption analysis of power grid distribution transformers based on an improved genetic algorithm

Yubin Lin, Jiyu Li, Xiaofei Ruan, Xiaoyu Huang and Jinbo Zhang

Department of Evaluation Center, Economic and Technological Research Institute of State Grid Fujian Electric Power Co., Ltd., Fuzhou, Fujian, China

## ABSTRACT

With the promotion of energy transformation, the utilization ratio of electrical power is progressively rising. Since electrical power is challenging to store, real-time production and consumption become imperative, imposing significant demands on the dependability and operational efficiency of electrical power apparatus. Suppose the load distribution among multiple transformers within a transformer network exhibits inequality. In such instances, it will amplify the total energy consumption during the voltage conversion process, and local, long-term high-load transformer networks become more susceptible to failures. In this article, we scrutinize the matter of transformer energy utilization in the context of electricity transmission within grid systems. We propose a methodology grounded on genetic algorithms to optimize transformer energy usage by dynamically redistributing loads among diverse transformers based on their operational status monitoring. In our experimentation, we employed three distinct approaches to enhance energy efficiency. The experimental findings evince that this approach facilitates swifter attainment of the optimal power level and diminishes the overall energy consumption during transformer operation. Moreover, it exhibits a heightened responsiveness to fluctuations in power demand from the electrical grid. Experimental results manifest that this technique can truncate monitoring time by 27% and curtail the overall energy consumption of the distribution transformer network by 11.81%. Lastly, we deliberate upon the potential applications of genetic algorithms in the realm of power equipment management and energy optimization issues.

## INTRODUCTION

With the development of the economy, people's demand for electricity has been increasing year by year, and various industries have become increasingly dependent on electricity. However, current technology cannot store large amounts of electrical energy due to technical limitations, storage costs, reliability and safety factors. This leads to real-time production and consumption of electricity. At the same time, this characteristic further limits each link in the power system to maintain continuity. Therefore, given the unique properties of electrical energy, providing safe, reliable and economical high-quality

Corresponding author
Yubin Lin,
18606932711@163.com

electrical energy for customers is an urgent problem that needs to be solved in the power system (*Fayazi et al., 2023*).

The generation and utilization of electrical energy often occur at remote locations, necessitating the establishment of secure and dependable means of energy transmission. The predominant approach employed for power transmission is high-voltage alternating current (HVAC). While HVAC technology is a well-established, steadfast, and broadly applicable transmission method, it is not without its drawbacks. For instance, over long distances and with high capacity, HVAC transmission can lead to considerable energy dissipation and diminished efficacy (*Mirza & Gupta, 2022*). In order to mitigate energy loss during transmission, transformers elevate the voltage of the power prior to transmission, generating high-voltage electricity. Transformers also serve the purpose of reducing the voltage of the power before its utilization at the receiving end.

Distribution transformers, or mutual inductors, are crucial equipment in power grids. They facilitate the transfer of electrical energy between generators, substations, cable lines, and other electrical equipment. Transformers adjust the input voltage to appropriate load levels, which ensures system safety, stability, and reliable operation. As power systems evolve and upgrade, distribution transformers undergo continuous technological innovation and improvement to meet growing electricity demands and environmental requirements (*Thiviyanathan et al., 2022*).

However, when multiple transformers operate together, if the load distribution is uneven, it will cause an increase in the overall energy consumption of transformer operation, resulting in energy waste. According to statistical data, the energy consumption of transformers is an essential component of the entire power system's energy consumption. Overloading of distribution transformers will reduce their conversion efficiency. If the load on the transformer network is uneven, it will increase energy consumption and may negatively affect the power supply from the grid (*Elsisi et al., 2022*). Therefore, it is necessary to dynamically allocate the load of each transformer through effective optimization methods to improve energy transmission efficiency. Transformers are also a vulnerable component in the power system. Due to the large number of components inside transformers, faults often occur. Once a transformer fails, it will cause power supply interruption and inability to meet load power requirements in the power system. Long-term uneven distribution of loads and long-term high-load operation of local equipment will accelerate the wear and tear of transformer equipment, leading to an increase in failure rate (*Medina et al., 2022*). Therefore, optimizing the distribution of transformer loads and promptly detecting and resolving transformer faults is crucial.

By monitoring transformer operation conditions and optimizing load distribution among transformers, these problems can be resolved and the overall energy efficiency and stability of the entire power system can be increased (*Nousdilis, Christoforidis & Papagiannis, 2018*). Currently, more research is exploring reliable transformer operation monitoring and load optimization algorithms. One direction is power consumption optimization of distribution transformers based on various genetic algorithms.

In this article, we propose a method using the Improved Adaptive Genetic Algorithm (IAGA) to optimize the energy consumption of distribution transformers in power grids,

which can effectively avoid transformer faults and uneven load distribution problems. Specifically, this method first requires real-time monitoring of the state of transformers, including voltage, current, temperature and other parameters. Then, an improved genetic algorithm is used to optimize the load distribution of each transformer so that the overall system energy consumption is minimized. Finally, based on the optimized load allocation scheme, control the loads of each transformer to achieve optimal energy transfer efficiency.

This paper amalgamates the energy utilization challenge pertaining to distribution transformers, presenting an enhanced adaptive genetic algorithm for the optimization of transformer operations. The experimental results show that after application, this method can reduce monitoring time by 27% and decrease overall energy consumption in the distribution network by 12%, effectively reducing energy consumption and improving energy transfer efficiency. At the same time, this method can also help power system managers monitor the real-time operational status of transformers and timely detect faults and take corresponding measures for repair to ensure the safe and stable operation of the entire power grid.

The main contributions of this article are summarized as follows:

1) To assist managers in quickly identifying possible problems from unbalanced loads, a distribution transformer energy consumption monitoring system has been developed.
2) A power distribution transformer energy optimization model based on an improved genetic algorithm was proposed, which can reduce the overall energy consumption of the distribution transformer network and lower the occurrence rate of faults.
3) The prospects of genetic algorithms in the field of power technology and large-scale equipment networks are analyzed and discussed, and it is believed that the application of intelligent optimization methods, such as genetic algorithms, is the future direction compared to traditional optimization methods.

## RELATED WORKS

Based on the research of Fraser and Bremermann, Holland first adopted the encoding method of binary strings to implement the operations of crossover and mutation operators and proposed the main idea of the genetic algorithm (*Holland, 1984*). Genetic algorithm is a global search evolutionary computing method based on the idea of evolution.

It operates on a group of binary strings (called chromosomes or individuals) known as the population. Each chromosome corresponds to a solution to the problem being solved (*Sumida et al., 1990*).

In genetic algorithms, crossover probability $P_c$ and mutation rate $P_m$ affect the algorithm's effectiveness. In simple genetic algorithms, $P_c$ and $P_m$ are fixed values. A larger $P_c$ leads to a faster generation of new individuals but also increases the possibility of destroying high-fit individuals. The mutation probability $P_m$ is too tiny to produce unique individuals. Later, Srinivas et al. came up with the idea that the crossover probability and variance probability could be automatically adjusted when the fitness of individuals in the

population converge or disperse, which is the AGA (*Srinivas & Patnaik, 1994*). However, this method is only effective when the population is at a later stage of evolution.

Considering the optimization effect of the genetic algorithm, it has been used early on in solving optimization problems in the power system. *Chen & Cherng (1999)* proposed an effective method to optimize the phase arrangement of a primary feeder distribution transformer to improve system imbalance, reduce losses and achieve good results. Genetic algorithms can also be combined with neural networks to solve optimization problems. *Georgilakis et al. (2001)* proposed a method based on combining neural networks and genetic algorithms to reduce losses. Compared with previous methods, transformer iron loss was significantly reduced, thus saving important economic costs for transformer manufacturers.

However, these studies were relatively early, and since then, there have been new developments in the application of genetic algorithms in the power industry. *Piasson et al. (2016)* proposed a multi-objective model based on a genetic algorithm for optimizing maintenance planning in distribution systems with reliability as its focus on addressing fault problems. Experimental results showed that this method has good robustness and high quality in EPDS maintenance planning, although the optimization variables are relatively singular. Later, with the development of hardware, various sensor devices have been used to monitor the real-time operation status of power systems. *Jaiswal et al. (2018)* introduced an intelligent online state monitoring system for analyzing distribution transformer status. The system comprises intelligent electronic devices that can receive transformer inputs through sensors, providing a smart management solution for the increasingly growing distribution network. However, this method has a strong hardware dependency and many monitoring devices increase energy consumption and costs to some extent. In addition, *Foltyn et al. (2021)* studied the problem of power loss in power production and transmission equipment about energy transformation issues, proposing an optimization method based on improved genetic algorithms to reduce the energy consumption of transformer equipment and minimize operating costs for the grid. *Zhou et al. (2020)* successfully applied multiple hybrid models based on genetic algorithms to predict the hourly output power of photovoltaic systems using machine learning techniques. *Lorencin et al. (2019)* proposed a genetic algorithm-based multilayer perceptron for estimating the power output of a combined cycle power plant. These methods have shown improved performance compared to previous optimization methods. But as the scale of the power grid expands, there are many parameters for transformer-free optimization, with a large range of values and multiple optimization goals. These methods require a large amount of computation, making it difficult to achieve good results in both optimization speed and accuracy when computing power is limited.

The above studies suggest that genetic algorithms hold substantial utility in the realm of power systems but still confront challenges requiring resolution. This article introduces a load optimization approach for distribution transformers founded on an enhanced adaptive genetic algorithm, yielding commendable optimization speed and average energy consumption outcomes.

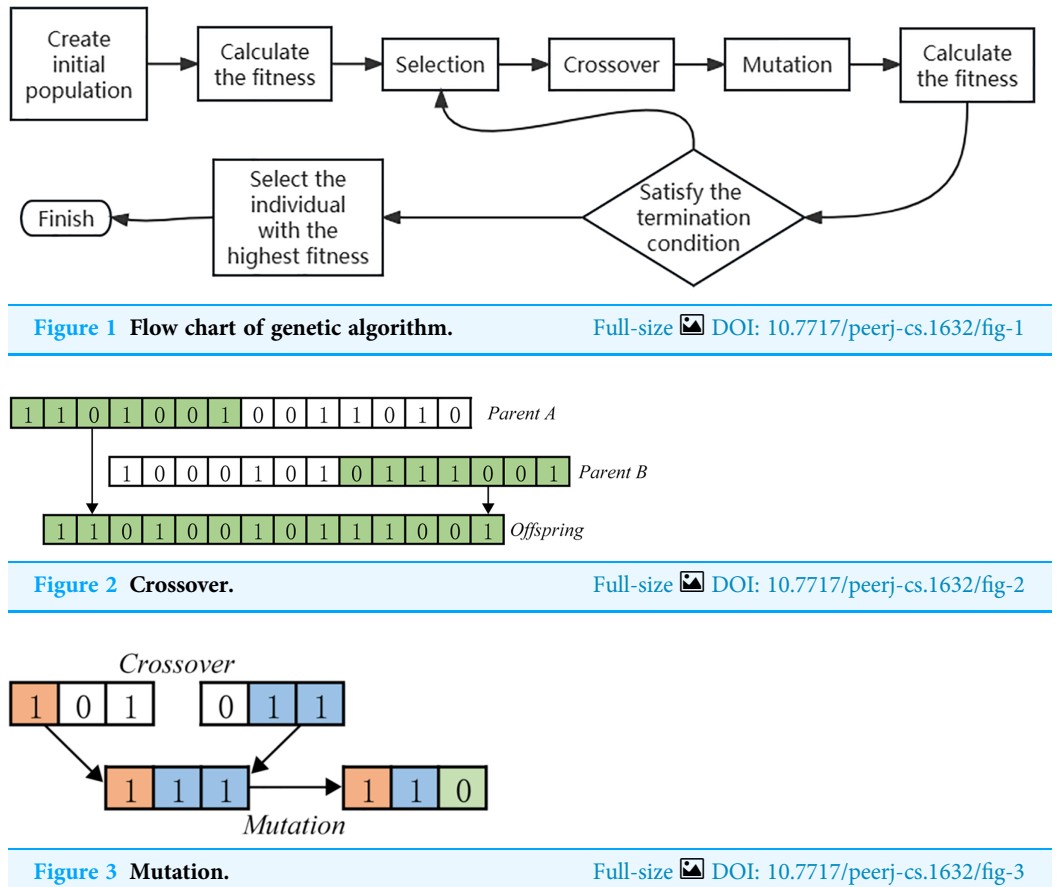

**Figure 1 Flow chart of genetic algorithm.**

**Figure 2 Crossover.**

**Figure 3 Mutation.**

## METHODS

The optimization process of a genetic algorithm is an iterative process in which the basic features of each generation are inherited by the next generation through genetic mechanisms (*Shukla, Pandey & Mehrotra, 2015*). The Simple Genetic Algorithm (Standard Genetic Algorithm) is a unified and fundamental genetic algorithm. Its process is shown in Fig. 1.

Basic genetic algorithms only use three basic genetic operators: selection, crossover, and mutation (*Karakatič & Podgorelec, 2015*), as shown in Figs. 2 and 3.

Genetic algorithms abstract the evolutionary processes of organisms, creating a search algorithm characterized by "generation + validation." These algorithms fully emulate natural selection and genetic mechanisms. Leveraging computer simulations of biological genetics and evolutionary processes, genetic algorithms bestow exceptional adaptive and optimization capabilities upon various artificial systems (*Sun et al., 2020*).

This article focuses on the energy consumption analysis of power grid distribution transformers. It proposes a method based on an improved genetic algorithm for dynamically allocating transformer loads, thereby optimizing the overall energy consumption of transformer operations.

The main problems with simple genetic algorithms are slow convergence speed and the tendency to produce premature convergence (*Sastry, Goldberg & Kendall, 2005*).

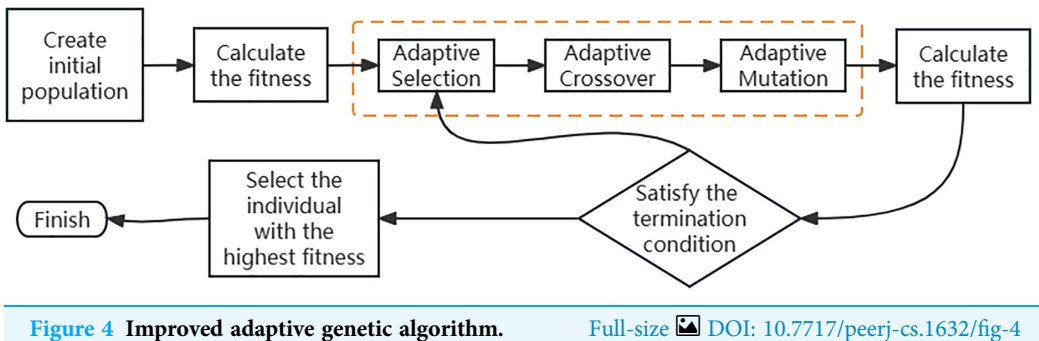

**Figure 4 Improved adaptive genetic algorithm.**

Making certain improvements to the basic genetic algorithm or combining it with other algorithms is an effective method for improving its efficiency and solution quality. Therefore, an adaptive genetic algorithm that automatically adjusts crossover and mutation probabilities based on fitness has been developed.

Adaptive genetic algorithm is an improvement of the basic genetic algorithm, which combines adaptive algorithms and genetic algorithms (*Deb & Beyer, 2001*). An adaptive algorithm is a fundamental algorithm that requires the algorithm to adapt to changes in certain limiting conditions during its execution. As shown in Fig. 4, an adaptive genetic algorithm replaces the genetic part of simple genetic algorithms with adaptive genetic operations.

In the fitness evaluation stage, an improved best-keeping strategy was applied in this article. The specific process is to compare the highest fitness of the newly generated population with that of the previous generation. If it is lower, a randomly selected individual from the new generation will be eliminated and replaced by the individual from the last generation with the highest fitness.

An improved optimal preservation strategy can ensure that the current optimal individuals will not be disrupted by genetic operations such as crossover and mutation (*Gao, Sun & Zheng, 2015*). Through experimental analysis, the relationship between the number of individuals in a population and the optimal number of individuals to be retained can be determined using the following method. When selecting an optimal number of individuals to maintain that does not exceed 10% of the total population size, a good balance point can be found in terms of optimization level and global search ability. This strategy is adopted in this article to retain the best individuals.

The general mathematical model for overall system optimization is a constrained optimization model, such as Formula (1).

$$\begin{cases} minF(x) \\ s.t.g_i(x) < g_i \end{cases} \tag{1}$$

The first formula represents the optimization objective, while the second formula represents the constraints on the model.

Its objective function F is shown in Formula (2).

$$minF = \sum_{i}^{N}(1 - c_i)P \tag{2}$$

where F is the overall energy consumption, $c_i$ is the transformer conversion efficiency of the $i$-th distribution transformer, and P is the input power of the transformer.

The genetic algorithm is an unconstrained optimization algorithm, while there are constraints in the grid, for which the penalty function method is used to deal with, as shown in Formula (3).

$$P(x) = F(x) + \sum_{i=1}^{n} f_i(x) \tag{3}$$

$$f_i(x) = \begin{cases} 0 \\ a_i \dfrac{g_i(x) - g_i}{g_i} \end{cases} \tag{4}$$

A penalty function transforms a nonlinearly constrained optimization problem into an unconstrained optimization problem with penalties for violating the constraints. $P(x)$ is the adjusted objective function after modification. Where, $f_i(x)$ is the penalty function, $i =$ 1, 2, 3, …, n.

As the optimization process nears its end, the genetic algorithm gradually converges because the fitness values of individuals in the population are relatively close. Therefore, it becomes difficult for the algorithm to continue optimizing and selecting, resulting in oscillations around the optimal solution. At this point, a fitness scaling method is used to amplify individual fitness values and improve selection performance. This article uses a Formula (5) as the fitness calculation formula.

$$f' = \frac{1}{f_{\max} + f_{\min} + \delta}(f + |f_{\min}|) \tag{5}$$

In the equation, $f'$ is the calibrated fitness; $f_{\max}$ is an upper limit of $f$; $f_{\min}$ is the lower limit of $f$; taking $\delta$ aims to prevent division by zero and increase the randomness of the genetic algorithm, where $\delta$ is a natural number within (0,1); $f$ is the original fitness before calibration; and $|f_{\min}|$ Ensures that the calibrated fitness will not be negative.

## EXPERIMENT

For the convenience of calculation, this article takes a scheduling period of 1 h during simulation calculations. It assumes that the input power of the power grid is relatively stable and will not change dramatically.

First of all, we need to initialize the population. The number of the population M is 40. Under the initial conditions, the overall power of the distribution transformer is approximately $1.79 \times 10^4$ W. Then, three genetic algorithms are used to optimize the power consumption of the distribution transformer network. As the load is adjusted, it is expected that the overall energy consumption of the network will continue to decrease during experimentation. Figure 5 shows the results of the improved adaptive genetic algorithm (IAGA) proposed in this article compared with the adaptive genetic algorithm (AGA) and simple genetic algorithm (SGA) optimization iterative process.

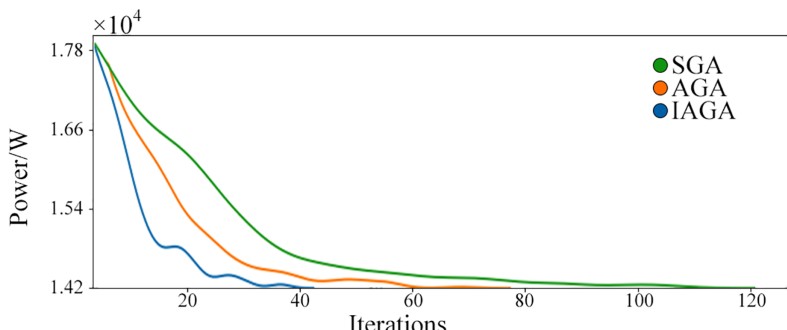

**Figure 5** The change of power with iterations.

**Table 1 Iteration times and average energy consumption.**

| Method | Average | Iterations |
|---|---|---|
| SGA | 1.6127 | ≈109 |
| AGA | 1.5515 | ≈62 |
| IAGA | 1.4224 | ≈42 |

As shown in Fig. 5, after optimizing the variable voltage network load distribution using the genetic algorithm, the overall energy consumption of the variable voltage network decreases continuously with the iterations of the algorithm. The optimization speed of different algorithms is different, and IAGA has the best result and the energy consumption value stabilizes after about 40 iterations and achieves a better result. The SGA optimization is the slowest, falling into the same local optimum problem as the AGA algorithm at 43 iterations. Still, as the number of genetic iterations increases, the overall power under the SGA method decreases slowly until it reaches a result similar to IAGA after about 109 iterations.

Table 1 shows the average energy consumption and number of iterations required to achieve lower energy consumption during the optimization process of a transformer network over 109 h using three different methods. As each iteration in the experiment was fixed at 1 h, the number of iterations also represents the time required for optimizing the transformer network. From the table, it can be seen that IAGA has better performance compared to other methods. Looking at the average energy consumption during optimization, faster attainment of optimal power levels results in lower overall energy consumption during transformer operation and better ability to respond to changes in power demand from electrical grids. The optimization time for IAGA is only 38.53% that of SGA's, resulting in an 11.81% reduction in energy usage when applied to single-stable power variation load optimization processes for distribution transformers as calculated over 109 h.

Due to the randomness involved in the optimization process of genetic algorithms, there may be differences in energy-saving effects in actual power system applications. As illustrated in Fig. 6, the algorithm put forth within the confines of this scholarly work has

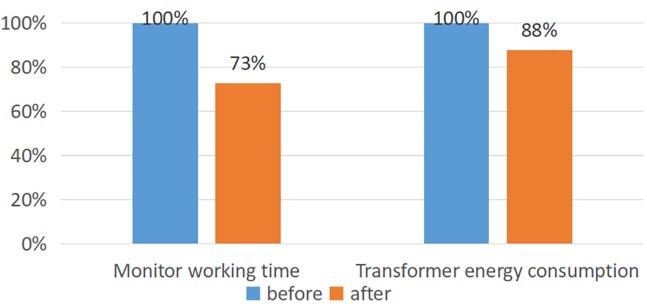

**Figure 6 Comparison of application.** This method can reduce monitoring work time by 27% and overall energy consumption of distribution transformer networks by 12%.

been adeptly integrated into a power distribution transformer energy monitoring system. Its purpose lies in assisting the managerial cadre in promptly discerning latent hazards of load imbalance. This, in turn, culminates in an impressive average curtailment of 27% in the time dedicated to monitoring endeavors, coupled with a commendable decrease of 12% in the overall energy consumption of the power distribution transformer network.

## DISCUSSION

The current enterprises are facing a complex problem in designing a low energy consumption distribution transformer network and reducing the failure rate of transformers, as market competition becomes increasingly fierce and energy saving and consumption reduction take center stage. The dispatching operation of grid transformers presents new opportunities and challenges in this evolving industry (*Zhou et al., 2016*). It is crucial to address the high losses in distribution transformers, which account for significant losses in power systems. By optimizing the dispatching strategy of distribution transformers, enterprises can reduce the operation cost of electric energy production, enhance the operation performance and stability of the power grid, and minimize losses during transformer operation. These improvements yield evident social and economic benefits (*Rao & Zhang, 2020*). The traditional transformer dispatching network lacks monitoring capabilities that can respond quickly, the efficiency of maintenance and inspection is low, and it is impossible to determine whether the minimum energy consumption has been reached even if the operation is stable (*Syed et al., 2021*). Thus, it seems that the use of optimization algorithms for transformer energy consumption has become an inevitable trend.

The experimental outcomes presented in this article demonstrate that the enhanced adaptive genetic algorithm exhibits superior global optimization capabilities and solution accuracy. It has been established as an effective approach for addressing the energy consumption optimization challenges associated with power transformers. The comprehensive energy efficiency cost of the transformer is significantly reduced, and the socio-economic benefits of the product are improved. Improvements such as fitness transformation, selection probability, crossover probability and variation probability that dynamically change according to individual fitness values are adopted in the process of

design calculation, which play an active and obvious role in the process of optimization, thus providing a meaningful exploration of the optimal design of transformers, indicating that the adaptive genetic algorithm has specific use value for the optimal design of transformers, and providing a direction for the genetic algorithm in the engineering optimization The application of genetic algorithm in the field of engineering optimization is indicated.

The energy consumption optimization of distribution transformers must consider technical performance index constraints, process technology index constraints, and material performance constraints (*Das et al., 2018*). Although the work in this stage has achieved relatively satisfactory research results, there is still room for energy consumption optimization in other stages.

## CONCLUSION

This article focuses on the distribution transformer load optimization analysis based on an improved adaptive genetic algorithm. The mathematical model of distribution transformer load distribution is established, and the constraints of distribution transformer load demand are considered. The improved adaptive genetic algorithm is proposed, and the selected model is solved numerically. The effectiveness of the proposed algorithm is verified with arithmetic examples, which provide a valuable reference for the economic dispatch of micropower sources in the microgrid and the application of the genetic algorithm in the grid. Finally, the prospect of applying genetic algorithms in power technology and large-scale equipment networks is analyzed and discussed.

One of the key innovative contributions of this research is the development of an improved adaptive genetic algorithm, which enhances the optimization performance compared to traditional approaches. The algorithm adapts and evolves over time by incorporating adaptive strategies, leading to more efficient and accurate solutions. This innovation enables the algorithm to handle the complex optimization problem of transformer energy consumption effectively. The numerical examples presented in this article validate the effectiveness of the proposed algorithm. The results serve as a valuable reference for the economic dispatch of micro power sources in microgrids. Furthermore, the successful application of the genetic algorithm in this context highlights its potential for broader application in power technology and large-scale equipment networks.

In conclusion, the genetic algorithm demonstrates remarkable effectiveness and feasibility in addressing the optimization problem of transformer energy consumption within the scope of this study, thus highlighting its immense potential for practical applications. However, it is crucial to emphasize that achieving exceptional optimization outcomes when employing the genetic algorithm for transformer energy consumption optimization necessitates carefully configuring parameters. Moving forward, it is recommended to explore the application of the genetic algorithm to more intricate transformer energy consumption optimization problems, aiming to achieve even greater optimization efficacy.

## Funding

The authors received no funding for this work.

## Competing Interests

All the authors are employed by Economic and Technological Research Institute of State Grid Fujian Electric Power Co., Ltd.

## Author Contributions

- Yubin Lin conceived and designed the experiments, performed the experiments, prepared figures and/or tables, and approved the final draft.
- Jiyu Li conceived and designed the experiments, performed the experiments, authored or reviewed drafts of the article, and approved the final draft.
- Xiaofei Ruan performed the experiments, analyzed the data, prepared figures and/or tables, and approved the final draft.
- Xiaoyu Huang analyzed the data, performed the computation work, authored or reviewed drafts of the article, and approved the final draft.
- Jinbo Zhang analyzed the data, performed the computation work, prepared figures and/or tables, and approved the final draft.

## Data Availability

The data is available at Zenodo: Martinez-Ceseña, Eduardo, & Kong, Wangwei. (2022). Simplified electricity transmission system from UK (2020) [Data set]. https://doi.org/10.25747/z85s-se29.

## Supplemental Information

Supplemental information for this article can be found online at http://dx.doi.org/10.7717/peerj-cs.1632#supplemental-information.

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
