# Peer review of "Energy consumption analysis of power grid distribution transformers based on an improved genetic algorithm"

_PeerJ Computer Science, doi:10.7717/peerj-cs.1632_

## Round 0.1 · original submission · Major Revisions

Dear authors

Your paper has been checked by the experts in their field and you will see that they have a couple of improvement suggestions, which can uplift the quality of your manuscript.

Please also improve the quality of the article by professional writing, revise the abstract by adding concrete results.

Please revise the paper and resubmit.

**Language Note:** The Academic Editor has identified that the English language must be improved. PeerJ can provide language editing services - please contact us at copyediting@peerj.com for pricing (be sure to provide your manuscript number and title). Alternatively, you should make your own arrangements to improve the language quality and provide details in your response letter. – PeerJ Staff

Reviewer 1 ·

Basic reporting

Due to electric energy is difficult to store, it needs to be produced and consumed in real-time, which puts high requirements on the reliability and efficiency of power equipment. This article analyze the problem of transformer energy consumption in the transmission process of power grid system, and propose a method based on genetic algorithm.

Experimental design

1. The description of experimental results in the abstract is too small to highlight the advantages of the method;
2. In the second section, the latest research on genetic algorithms can be arranged in chronological order;
3. The design system in the contribution is not introduced much in the paper;
4. Compared with traditional optimization methods, what are the core outstanding advantages of genetic algorithm reflected?
5. The descriptions of formulas (1), (3) and (4) are not enough to explain, and need to be supplemented here;
6. The introduction paragraph of the method section is too messy for readers to understand, and the author can divide it into the introduction before and after optimization;
7. Figure 5 compares three different algorithms, but it is monotonous and simple, and the author needs to add other different algorithms for comparison;
8. The experiment section is insufficient to support the innovation. I would suggest the author to add ablation experiments;

Validity of the findings

1. The discussion section is a bit like an introduction to the background, the author should emphasize the problem and how to solve it;
2. The description of the conclusion section is not enough, so it is necessary to strengthen the summary of the innovation points of this paper.

Additional comments

The experimental results show that this method can reduce the monitoring time and reduce the network energy consumption, Alongside good things in the article there are still some shortcomings in this paper which should be added to improve the quality of the manuscript.

Reviewer 2 ·

Basic reporting

With the promotion of energy transformation, the consumption rate of electric energy is gradually increasing. Because electricity energy is difficult to store, it needs to be produced and consumed in real time. In this paper, the authors have analyze the problem of transformer energy consumption in the transmission process of power grid system, and they proposes a method based on genetic algorithm to optimize transformer energy consumption by dynamically distributing load on the basis of monitoring different transformer operating states.

Experimental design

Overall the article seems to be fine but In addition, the following changes are required to be incorporated for this article:
1. Too few keywords are not enough to summarize the general content of the abstract;
2. The introduction section seems a little redundant, the author should focus on the topic of the article description;
3. The basic genetic algorithm in Figure 1 could use some descriptive introduction;
4. The innovation in Figure 4 is to replace the simple genetic algorithm with an adaptive genetic algorithm. What are the outstanding advantages of doing this?
5. The optimization time of the experiment section is a simple explanation, and I suggest the author can make a table to compare the explanation;
7. The experiment section lacks direct comparison of results, such as the accuracy rate, so the author needs to add related experiments;

Validity of the findings

1. The logic of the conclusion is not strong enough to summarize the main point. The author should rewrite the conclusion to strengthen the logical expression;
2. more details are required in the discusion section supported with evindences
3. I suggest the author add some excellent journal articles from recent years.

Additional comments

In this paper the genetic algorithm has high optimization effect and feasibility in transformer
energy consumption optimization problem, and has wide application prospect, so the contribution is decent and clear

---

## Round 0.2 · accepted · Accept

Dear Authors,

Based on the input received from the experts in the field regarding your article, I am pleased to inform you that we are accepting your paper for publication.

Reviewer 1 ·

Basic reporting

All the sections are improved as per suggestion.

Experimental design

The experimental section is also improved.

Validity of the findings

This manuscript is the best contribution to the research.

Reviewer 2 ·

Basic reporting

All the comments have been addressed by the author, I don't have any further comments. The paper looks in a very good standard and I am happy to accept the paper for publication in Peer J computer science journal.

Experimental design

.

Validity of the findings

.